# South Tyrol Healthcare Company: A Look at a Peculiar Model of Claims Management in Italy and Analysis of Its Last 11 Years

**DOI:** 10.3390/healthcare12111070

**Published:** 2024-05-24

**Authors:** Martina Zanon, Francesco Randazzo, Valter Equisetto, Paolo Martini, Silvia Winkler, Christian Leuprecht, Stephan Vale, Antonia Tessadri, Monica Concato, Stefano D’Errico

**Affiliations:** 1Servizio di Medicina Legale Aziendale, Azienda Sanitaria dell’Alto Adige, 39100 Bolzano, Italy; medleg.bz@asdaa.it (M.Z.); valter.equisetto@sabes.it (V.E.); 2Ufficio affari generali e assicurazioni, Azienda Sanitaria dell’Alto Adige, 39100 Bolzano, Italy; uagbz@asdaa.it; 3Conciliation Commission of South Tirol, 39100 Bolzano, Italy; concilmed@provincia.bz.it (S.W.); christian.leuprecht@provinz.bz.it (C.L.); stephan.vale@sabes.it (S.V.); a.tessadri@inail.it (A.T.); 4Department of Medicine, Surgery and Health, University of Trieste, 34127 Trieste, Italy; monica.concato@studenti.units.it

**Keywords:** alternative dispute resolution solutions, clinical risk management, medical liability, conciliation and compensation in medical claims

## Abstract

Background: Alternative Disputes Resolution (ADR) systems are becoming increasingly important tools in recent years for the management and resolution of health responsibility cases, but their dissemination and efficiency are still poorly described. The purpose of this paper is to present an ADR system in the autonomous province of Bolzano: the Conciliation Commission. Methods: systematic collection of all claims of the South Tyrol Sanitary Service that were dealt with in the Conciliation Commission from 1 January 2012 to 31 December 2022. Results: closing times of the applications received turn out to be less than a year in 63.8% of the cases, with a number of cases managed rather stably in the time, even if minimal if compared to the number of complaints received to the South Tirol Health Service. Only 5.3% of the application continued the legal process before a civil court. Conclusions: the Conciliation Commission of South Tirol appears to be an excellent instrument for the resolution of disputes in the healthcare field, with rapid resolution times and little to zero costs for the healthcare company, a public health institution. Despite its effectiveness, it seems to be a tool that is still little-known in South Tyrol.

## 1. Introduction

Alternative Disputes Resolution (ADR) systems in medical liability are nowadays important instruments for out-of-court dispute settlements to avoid litigation and help resolve conflicts between patients and healthcare providers/hospitals/health services [1]. The purpose of these systems is to limit access to the courtrooms, to contain the costs of justice in addition to the timing of the resolution of a dispute, and, moreover, to have a pacifying function in order to mitigate the growing conflict between patients and hospitals/doctors [2,3,4]. The importance of ADR was even recognized by the European Commission, which promotes the use of ADR for dispute settlements [5].

ADR systems are becoming increasingly widespread worldwide. Some examples are the USA, where the use of ADR tools is still rather fragmented among the states but recollection of these instruments is becoming increasingly common. In some states, such as Wisconsin, North Carolina, or Pennsylvania, their use is regulated by law and ADR is required before litigation. It also seems that in the last decade, there has been a political impetus for ADR [6]. Japan has established the Japan Medical Association, which provides medical liability insurance to both members and non-members and conducts inquiries into claims to assess whether or not there is negligence. Negotiation takes place in the case of evidence of negligence, and the medical association and insurance company will cooperate to resolve the dispute in accordance with the decision of the Liability Review Committee in order to ensure a fair compensation [7,8]. In the UK, hospital committees investigate cases of complaints using a standardized National Health Service (NHS) protocol. If the patient is not satisfied with its outcome, they can proceed to the Parliamentary and Health Service Ombudsman (PHSO), which is independent of the NHS and judges the complaint without the need for a trial [9]. In China, the “Medical Disputes Prevention and Treatment Regulations” of 2018 emphasize the resolution of medical disputes through litigation or, where ADR methods exits, through mediation or arbitration. The use of mediation has thus arisen in recent years. In the case of an agreement, a trial cannot be initiated [10,11].

Other examples are the Fonds des accidents médicaux (FAM), a service within the National Institute for Health Insurance and Invalidity (NIHDI), founded in 2010 in Belgium, which compensates the victims of healthcare injuries through public funding [12]; the dispute committees in the Netherlands, founded in 2016 and compulsory in cases of complaints to the healthcare provider/healthcare organization before any legal action is taken [13,14,15]; the Conciliation and Compensation Commission for Medical Accidents (CCI) in France, a part of the National Office for Compensation for Medical Accidents, Iatrogenic Diseases, and Nosocomial Infections (ONIAM), a public institution established by the Law in March 2002 to compensate damages attributable to medical research activities, iatrogenic injuries, nosocomial infections and, as of 2020, from vaccination against SARS-CoV-2 [16,17]; the Conciliation Commissions in Austria, which are committees set up in each Physician’s Order to assess medical liability cases in order to avoid in court legal action [18,19,20,21,22,23,24,25]; the evaluation and conciliation committees in Germany, which are very similar to those in Austria, and offer the possibility to request a free opinion from a committee of doctors and lawyers, to both a citizen who thinks they have suffered damage and to health professionals, for more than 40 years [26,27].

In Italy, the use of ADR tools in terms of health liability is currently only partial, and is among the main reasons for a fear of incurring an administrative accounting liability judgment for public employees who authorize a conciliatory payment. To remedy this problem, the Law n. 206 of 26 November 2021 for the simplification and rationalization of judicial disputes, in letter g of article 1 paragraph 4, provides that “*for the representatives of the public administrations referred to in article 1, paragraph 2, of the legislative decree 30 March 2001, n. 165, that conciliation in the mediation procedure or in court does not give rise to accounting liability, except in cases where there is willful misconduct or gross negligence, consisting in inexcusable negligence deriving from the serious violation of the law or from the misrepresentation of the facts*”.

From summer 2007 in South Tirol, an autonomous provincial body was made available to the population by the Public Administration to facilitate reconciliation between patients and health services in cases of health disputes. Established by the Provincial Law No. 7 of 5 March 2001 and its subsequent amendments, the Conciliation Commission represents an ADR institution competent for all cases in which a patient considers that his or her health has been harmed by an omission or action of a healthcare provider, both for public and private institutions. It is also competent for all cases in which a patient considers that his or her health has been harmed as a result of omitted or incorrect information. Its absolute independence and impartiality in the evaluation of errors in the healthcare sector are due to its independent organization comprising two lawyers, one in the role of president, chosen from the Bar Association of Bolzano; one doctor with specialization in forensic medicine, who has no professional relationship with the provincial health service; and one provincial employee acting in administration for the management of applications. 

The work of the commission is based on three principles: voluntariness of the procedure; free of charge procedure, except for the payment of the due stamp duty; and the non-binding nature of the medical–legal opinions expressed. Applications to the Conciliation Commission have to be made through a specific form, where the following must be indicated: the identification data of the applicant, if possible the name of the health professionals or structures involved, and a brief description of the fact with the explicit relevant complaints. Legal assistance or a technical opinion of the requesting party are not necessary. The application can also be presented by the health professional themself or the healthcare company to obtain an impartial evaluation of their work. Insurances of the doctors, respectively, in a public or private healthcare company may intervene in the proceedings before the commission.

The first phase of the procedure in the Conciliation Commission is the declaration of admissibility in the application, linked to the presence of all the formal requirements set out above and the absence of baseless assumption. Subsequently, a preliminary phase of study of the documentation is conducted. This helps to reduce the waiting times and to guide the discussion during the first hearing. Before the date of the first hearing, the parties are generally asked to prepare and submit a statement of position. During the first hearing, which is made in person, the parties take action and the commission carries out the function of guiding and coordinating the discussion in order to reach an out-of-court settlement. In the case of an agreement, which is signed by the parties, the insurance company of the health service or the healthcare professional is committed to paying the agreed sum without, however, acknowledging an actual fault of the doctors involved in the procedure.

In the case of no agreement, the parties may decide to move on to the next phase of the evaluation, which envisages the formulation of a reasoned technical opinion by the commission, with the possible use of an external technical opinion. The reasoned technical opinion is expressed in writing and then presented to the parties. However, this opinion is not binding and does not preclude recourse to the judicial trial. A summary scheme of the procedure before the Conciliation Commission is shown in Figure 1. 

The average cost of commission for the public administration of South Tirol in the last 10 years was approximately EUR 1600 per case [28,29,30]. 

The aim of this study is to present the system of the Conciliation Commission implemented in South Tyrol and also to evaluate the effectiveness of the Conciliation Commission as an ADR through a comparison with other analogue ADR systems in Europe and the available data on the judicial handling of health liability cases in Italy.

## 2. Materials and Methods

Using a Microsoft Excel electronic format, this study was conducted through a systematic collection of all claims in the South Tyrol Sanitary Service that were dealt with in the Conciliation Commission for dispute resolution from 1 January 2012 to 31 December 2022. In particular, data collection focused on several areas that can be summarized in 4 domains:Temporal: year of the claim; date of the event; date of the complaint; date of filing of the claim; dates and number of commission meetings; in the case of a request for external consultancy to the commission the dates relating to the assignment; expert operations and delivery of the final report; and, finally, the date of closure of the case for the commission and the company.Structure: area and operating unit involved; any legal and medico-legal assistance provided by the other party.Causal: patient complaint.Monetary: amount claimed by the counter party and amount foreseen by the Conciliation Commission.

Following the first phase of data collection and homogenization, the calculation functions available in Excel were applied and the results were finally summarized in graphic form and by means of summary tables.

## 3. Results

### 3.1. General Data

In 11 years, from 2012 to 2022, a total of 243 applications were received by the Conciliation Commission of the Autonomous Province of Bolzano, with the time trend summarized in Figure 2.

An application was made by the patient themself in 81.9% of the cases, while in the remaining 18.1% of the cases was made by a third party, usually a family member. Additionally, 55.1% of parties were male, while the remaining 44.9% were female. The age of the injured party, divided by decade, is summarized in Figure 3. In general, 57% of the claimants were between 41 and 70 years old at the time of the application.

Within the applications, there is the possibility of involving one’s own legal representative, as performed by the applicant in only 38.3% of cases. Of these, in 74.2% of the cases the representative involved was a lawyer, in 14% of the cases it was the ombudsman, and in the remaining 11.8% of cases a family member, doctor, or the ANMIC association were involved. Regarding identifications of the department and the doctor involved, identification has been mandatory in the application for many years. In 82.3% of the cases, the applicant directly stated the name of one or more of the doctors involved, and in 65% of the cases, of one or more involved departments. 

In the applications, 82.3% of the claimants indicated one department involved in the event of interest, 13.6% two departments, and 4.1% three departments. The involvement of the different specializations, broken down by area, is summarized in Figure 4.

Within the surgical area, orthopaedics and traumatology (46.2%) was the specialization most frequently called upon, followed at a distance by general surgery (16.7%) and gynaecology and obstetrics (9.1%). In the medical, area the most involved department was the emergency department (44.7%), followed slightly by the internal medicine department, which was involved in 9.4% of cases. Finally, in the area of services, applications concerned radiodiagnostics or anaesthesia and intensive care, respectively, in 36% and 32% of the cases, followed by the rehabilitation department (12%).

The complaints raised by the applicants were of a various kind, with a small number of applications (20 cases) concerning more than one complaint. The most frequent claim by the applicant was surgical error (83 cases), followed by therapeutic error (70 cases). Fewer, but still relevant in terms of number, were requests concerning diagnostic delays (34 cases) and diagnostic errors (25 cases). A summary of the allegations can be found in Figure 5.

### 3.2. Sequence of Events

The temporal distribution of claimed events can be seen from Figure 6, in which a peak in events is observed in the year 2012.

Given the date of the event and the application, it is possible to estimate the time elapsed between the two. The figure below (Figure 7) shows that 59.7% of the applications were received within 2 years of the event, 27.6% between 3 and 5 years, and 9.4% between 5 and 10 years. Furthermore, 3.3% of the applications were received more than 10 years after the event.

In detail, 72.8% of the applications received by the Conciliation Commission were processed and received an invitation for the first hearing within 6 months of the application, 14% waited a maximum of 1 year for the first hearing, while 2% had to wait almost 2 years. Additionally, 5% of the applications were assessed as inadmissible, so no in-person session was held, while in the remaining 1.6% of the cases, no session was held due to patient withdrawal (four cases) or due to agreement between the parties before the first hearing (two cases).

In 81% of the applications, a maximum of two face-to-face meetings were held; 5.8% of the cases involved three meetings; and in three cases, four or more meetings were required to complete the evaluation process. For 27 cases, there was no in-person hearing due to the inadmissibility of the application or withdrawal by the patient. Two cases are still ongoing.

With regard to the entire procedure, i.e., the total time between submission and closure, 42.4% of the cases were closed within 6 months, 21.4% between 6 months and 1 year, 29.2% between 1 and 2 years, 6.2% within 4 years, and 0.8% are still ongoing (Figure 8).

As for the duration of the commission’s operations, 97.1% of cases were closed within 2 years, while only a fraction were closed within 4 years; two cases are still ongoing (Figure 9). This means that the administrative management takes, on average, about 3 to 4 months of the total time of handling of a case in the Conciliation Commission.

The pandemic period seemed to have had little influence on the duration of the proceedings in the Conciliation Commission. The chart below shows that 36% of the cases dealt with during the pandemic period were closed within 1 year and 46% within 2 years. The remaining 18% were closed within 3 years (Figure 10). Comparing the case closure data summarised in Figure 8 allows us to state that the average closure time in the pandemic period was longer (+6 months) compared to the average of 10 months seen in the 2012–2023 period.

### 3.3. Handling of the Case

Only 39.5% of the cases presented to the Conciliation Commission were actually assessed by the commission, while 49.4% of the cases ended with a waiver of continuation of the assessment process or with a conciliation of the parties, which usually during the first hearing. A substantial number of cases, twenty-six, were assessed as inadmissible by the commission because of the absence of damage to health (six cases); because the 10-year limitation period had been exceeded (one case) in accordance with Article 18 (3) of the Presidential Decree no. 11 of 18 January 2007 (ten cases); or because the application had already been dealt with in another judicial context (three cases). One case is still being discussed at the moment without an assignment to the Conciliation Commission.

In 52% of the 96 cases assessed by the commission, an external opinion was requested. Over the years, apart from a peak in 2019, the number of external opinions has remained fairly stable, generally fluctuating between three and five per year.

### 3.4. Conclusion of the Proceeding

By the end of January 2024, 99.2% of the Conciliation Commission cases were closed. Of these, 26.7% were conciliated without requesting a written opinion of the commission, while in the 94 cases where an evaluation of the commission was requested, 31 cases (12.7%) had the liability of the healthcare provider acknowledged and 63 cases had a denied involvement of the healthcare professional. In 46 cases (18.9%), one or more parties to the proceedings decided to abandon the procedure. Of these, 71% were abandoned by the patient, 13% were abandoned by mutual agreement, and 16% were abandoned by the company or by the failure of the summoned healthcare provider to appear. Additionally, 6% of the cases were settled outside the commission (Figure 11).

In conciliated cases and in cases of acknowledged medical liability, the commission indicates the sum for which the parties have agreed or gives an indication of the compensation due for the event in its closing remarks. These sums are summarised in Figure 12. In only 38% of the total number of cases did the commission provide a compensation sum for the damages, in which 67% of cases amounted to less than EUR 10,000. The compensation provided was over EUR 50,000 for only 12% of the cases, with two cases valued at over EUR 1 million.

A quantification of the damage was received for 116 cases (47.7%), and it is theoretically possible to determine the difference between what the patient claimed economically and what was actually suggested from the commission. Unfortunately, two cases are still open and the commission did not assess the case in twenty-six cases, so this computation is only possible for the remaining eighty-eight claims received and assessed by the commission. In four cases (4.5%), the sum claimed was less than what was actually compensated, and in two cases, there was no difference between the request and the compensated sum. In the remaining 93.2% of cases, the sum claimed exceeded the compensation later awarded. The difference in the claim is very variable, with a peak (22.7%) between EUR 10,001 and 25,000, followed by claims with differences between EUR 0 and 10,000 (20.4%), between EUR 25,001 and 50,000 (15.9%), and between EUR 50.001 100.000 (12.5%). In one case, the difference exceeded EUR 1 million (Figure 12).

One of the objectives of the Conciliation Commission is to reduce litigation. Of the cases dealt with by the commission, only 14 decided to proceed through the courts. Of note, one case is currently in mediation and one case had gone to court before application to the commission.

In seven of these cases, the patient gave up the evaluation process, in one case it was the company that abandoned the process, in four cases the commission evaluated the case and denied the presence of responsibility on the part of the health company, two of these cases employed the aid of an external medical consultancy, and finally in two cases the commission evaluated the case with the aid of an external consultancy for one of the cases, concluding that there was responsibility on the part of the company. In one case of waiver on the part of the patient, the insurance company of the South Tirol Healthcare Service had made a financial proposal during the conciliation attempt, but the patient decided not to accept the proposal and to conclude the procedure in the Conciliation Commission without instructing the commission to assess the case (Figure 13).

Of the cases brought to litigation, 43% are currently still open, 28% closed with a liability verdict, 14% were open with a preventive technical assessment that ended with a positive verdict for the health authority, 7% were dismissed by the judge, and a further 7% settled before the preventive technical assessment. 

## 4. Discussion

The number of applications submitted to the Conciliation Commission appears to be rather stable in the years between 2012 and 2021, with an important deflection in the years 2020 and 2021 that is most likely linked to the pandemic period. The reasons for the deflection of applications in the year 2016, although minimal (−27%), are not known.

With regard to the applicants, the claimant corresponded with a patient in 81.9% of the cases, while in the remaining cases the person did not coincide because the application was signed by the parents of a minor (36%), or by the patient’s heirs in the event of death (43%) or because the patient was age very old (14%). In the remaining three cases, one was the administrator, one was a lawyer, and in the case of the last one the reasons that the mother applied instead of the adult child are not known.

As regards to department involvement, the greater involvement of the surgical area in litigation is not surprising, considering that both nationally and also in the South Tirol Health Service, surgical departments are those most affected by litigation. The involvement next of the medical area and then of the service area is an expected fact, considering both the analysis of corporate litigation from 2012 to 2021 in the South Tyrol healthcare company (in brief, ASDAA) and the data of the last MARSH report of 2023 [30] on healthcare litigation in Italy. Hence, the Conciliation Commission simply represents, in its own small way, a mirror image of corporate litigation. If we focus, instead, on the numerosity of the individual operating units, a comparison can be made with the national Italian data presented by the MARSH report of 2023 [30]. The top five positions in terms of number of disputes nationally are the same as those found in the Conciliation Commission (orthopaedics and traumatology, emergency department, general surgery, gynaecology and obstetrics and neurosurgery). Only in the sixth place do we find a difference in terms of the departments involved, with urology taking the place of otolaryngology, followed by radiodiagnostics instead of anaesthesia and intensive care. This makes it even clearer that the Conciliation Commission simply takes a snapshot of the national situation.

In relation to the complaints made by the patients, it is possible to compare what happens in the Conciliation Commission with national data. Nationally, the most reported event is the surgical error, as well as in the Conciliation Commission. In contrast to the national data, there are more claims for treatment errors than for diagnostic errors/delays in the Conciliation Commission, with a smaller difference in percentage terms (24% vs. 23%). To follow, we find a similar percentage of procedural errors (4.2% vs. 5% nationally), while there are far fewer requests for care-related infections in the commission compared to national data (3.2% vs. 5%). 

Moreover, it is interesting to point out that within the applications in the Conciliation Commission, theft/loss is not present, since it does not address any personal injury and, consequently, according to Art. 4bis of Provincial Law no. 7/2001, does not fall within the competence of the commission itself.

In terms of timing, 42.4% of the cases were closed within 6 months, 21.4% within 1 year, and another 29.2% within 2 years. If one considers that in Italy mediation lasts about 6 months, with most of the cases handled with a single session and at a conciliation rate of only 39%, and that litigation at first instance lasts about a year and a half, it can be said that the closure times of the Conciliation Commission are shorter than litigation and broadly similar to those of mediation, keeping in mind that the conciliation rate is higher [31,32,33]. A major impact on closure times is given by requests for external technical advice (52% of cases assessed by the commission and 20.1% of the total requests), which extends the time by at least 6 months to 1 year in relation to the time it takes to assign the task (average of 7 months) and deliver the work (average of 7 months after the assignment). The impact of the SARS-CoV-2 pandemic resulted in a slight increase in the time taken to close the cases (16 months vs. 10 months), but the closure time in the Conciliation Commission still remains below the time taken for a court case. However, it must be taken into account that the 18% of applications that were not assessed due to patient relinquishment and the 8.6% that were closed without even a face-to-face meeting due to inadmissibility of the application are also considered in the total time calculation. Clearly, if we were to disregard these applications, the closure times would be longer. All in all, the time taken by the Conciliation Commission in South Tyrol is fairly in line with the average duration of other alternative dispute resolution systems, making this a more-than-valid instrument for trying to resolve disputes in the health field without having to go to court.

A general assessment of the instrument represented by the Conciliation Commission in South Tyrol cannot fail to include a careful analysis of the conclusions of the process and the possible prosecution of the cases in the courts. First of all, it should be noted that 27% of cases were conciliated without an assessment by the commission, which essentially found itself having to act as a conciliator between the parties, who thus quickly reached an agreement (average closing time of less than 6 months). On the other hand, an equally large proportion of cases ended with a negative assessment by the commission, which denied the presence of any responsibility on the part of the healthcare staff involved. Of this quota, 6% of the cases unfortunately decided not to accept the commission’s assessment, which is totally impartial, and took legal action. Clearly, the explanation provided by the commission in these four cases, which was supported by the opinion of an external specialist in two cases, was not considered acceptable or sufficient, forcing the applicant to pursue the natural course of dispute resolution through the judicial route. It is interesting to note that out of these four cases, three are still open without any response from the preliminary technical assessment, while one has a negative assessment result, understood as a denial of responsibility on the part of the healthcare professionals involved. The presence of a double negative assessment by two completely autonomous bodies could help to close the case without the need for further meetings and could also help the patient to understand that the alleged error should perhaps not be considered as such. On the other hand, the commission recognised the presence of medical liability in 12.7% of the cases, in which the insurance company decided to comply with this decision and therefore provided compensation for the damage as stated by the commission. Surprisingly, it was not possible to reach an agreement between the claimant and the insurance company in two cases. Of these, one case is currently in court and the other is closed with an acknowledgement of the liability of the healthcare providers and a compensation of EUR 90,000 higher than what the Conciliation Commission had estimated (partly probably related to legal fees). In a very small proportion of cases (6%), the dispute after the first or second meeting in the commission was settled by an agreement between the parties outside the commission, which therefore did not assess the case. Of these, in one case a meeting never took place, with the parties reaching an agreement before the first hearing, and in another case, the commission was mandated to make an evaluation which never took place, as the parties reached an agreement before the external technical assessment could be carried out. A not-insignificant proportion of cases, however, waived the case evaluation, meaning that the patient, the company, or both parties, after an initial hearing, decided not to continue the process in the commission. Obviously, no justification for this choice is necessary, but the writer would like to try to make some assumptions. A first hypothesis could be the choice, once the company’s position had been heard and the explanation of the healthcare provider(s) involved had been heard, to forego the continuation of the evaluation process because the explanations provided were considered more than sufficient to explain what had happened and to justify the damaging event. A further hypothesis is that—especially in view of the two cases in which a conciliatory proposal had been made by the UNIQA insurance company, a third party liabilities insurer of the South Tyrol healthcare company—the claimant’s expectations of compensation were higher than the insurance company’s assessment of the damage, which relies on the evaluations provided by the Legal Medicine Service of ASDAA. Another hypothesis may be the choice, once the parties have been heard, to take direct legal action, since it was felt that the commission procedure might not be the most appropriate place to discuss the incident. This choice was made in 20% of the opt-out cases. It should be noted, however, that in two of these cases, the UNIQA insurance company had made a conciliatory offer that was rejected by the other party. The outcomes of these eight cases in court are still open for three cases, of which one case has a negative outcome by the preventive technical assessment. The five closed cases ended with a judgment of liability for three cases, one settlement during the process of investigation, and one rejection by the judge.

In order to assess the efficiency of the South Tyrolean Conciliation Commission, a comparison with other European realities with very similar tools and purposes could also be useful. Unfortunately, this comparison is not always easy because of the different collections and presentations of data. These differences make it impossible to compare the South Tirolean data with data from the Netherlands and Austria, while it is possible, albeit with limitations, to compare with other European countries that use dispute resolution systems very similar to the Conciliation Commission. One example is Belgium with the FAM, which deals with compensation in the event of health damage, or France, with its Commision de Concilition et d’Indemnisation des Accidents Mèdicaux, or Germany, with its conciliation commissions within the medical association. A comparison in the number of applications submitted is difficult because of the national distribution of these bodies, unlike the Conciliation Commission which only deals with cases occurring within the territory of the Province of Bolzano. On the other hand, it is interesting to note that in the case of Belgium, 14% of the applications are inadmissible, while in France 28% and in Germany this is up to 44%. The reasons for inadmissibility are various, from the time criterion (5 or 10 years’ interval between the event and the claim) to the completeness of the claim and participation of all the parties, to the need for a healthcare-related injury criteria that are not very different from those applied by the Conciliation Commission in South Tyrol, where, nevertheless, the inadmissibility rate is lower (8.6%), probably meaning that the inadmissibility criteria are less restrictive or that the population or the applicants lawyer are aware of these exclusion criteria and therefore do not use this provincial tool [34,35,36]. The South Tyrol Conciliation Commission has similar settlement times to those in France (7–10 months), demonstrating a better ability to process applications than Germany (15 months) and Belgium, where average closing times are very high indeed [35,36]. The data from Germany also allow us to compare the departments that are most involved, demonstrating the well-known trend towards greater involvement of the surgical area, particularly specialisations such as orthopaedics and traumatology and general surgery. It is interesting to note radiodiagnostics plays a very important role in the applications submitted in Germany, while it occupies a much lower position in the provincial area [36]. In the cases presented to the commission that were then actually assessed, 40% had a negative assessment outcome, a significantly lower percentage than in Germany (73%) and Belgium (72%). This clearly brings the number of cases in which the responsibility of the medical staff was recognised to a higher percentage for cases from the province of Bolzano (38%) compared to Belgium (14%), Germany (26%), and France (23-30%). Clearly, this difference is strongly influenced by the different legislative provisions on health responsibility between the various European countries, making one think of a greater tendency on the part of Italy to protect the patient, in the first place, in case of damage through the perspective of compensation rather than through other social systems [35,36,37].

Only for France was it possible to extrapolate the number of unsuccessful cases that went through a subsequent judicial procedure. This percentage is enormously higher (30% vs. 6%) than in South Tyrol, which is probably linked to the way the entire procedure is conducted. In fact, in South Tyrol, the sessions take place in person and the purpose is to make the patient and the healthcare company feel part of the proceeding, while in France there is only one meeting with the patient and the company involved, perhaps leading to a loss of trust on the part of the patient towards the commission and its work. It is not by chance that a study carried out in the Netherlands has shown a strong dissatisfaction of applicants with the local commissions, where the evaluation process, which mainly involves evaluation of the documentation attached to the application, very often leads to the denial of responsibility of the involved healthcare provider [38].

A limitation of this study is the difficulty of assessing the actual efficiency and effectiveness of the South Tyrol Conciliation Commission in the settlement of health professional liability disputes by comparison with other similar ADR systems around the world. This is due to the diversity of the tools, with different methods of case handling, but also to different methods of compensation recognition and data presentation.

The authors hope that this article will serve as a reference and as a stimulus for the publication of many more reports in which the work of an ADR system is presented extensively, so that we may be able to more clearly assess the actual effectiveness of each system and alleviate any shortcomings in the future.

## 5. Conclusions

When it comes to claim management, the South Tyrol Health Service has a tool that is unique in the country at its disposal, allowing it to resolve a significant percentage of cases through a form of Alternative Dispute Resolution, which often allows a case to be dealt with quickly. This tool, represented by the Conciliation Commission of the Autonomous Province of Bolzano, appears to be an excellent instrument for the resolution of disputes in the healthcare sector, with rapid resolution times, a zero cost for the healthcare company, and a fairly low cost for the public administration (about EUR 2000/case). Despite its functionality, it seems to be a tool that is still little-known in South Tyrol, since only a very small part of the claims that come to the attention of the South Tyrol Health Service have involved an application to this commission, demonstrating that there is a need to improve or perhaps create an advertising campaign. Moreover, settlement time could be improved within the limits clearly imposed by the administrative management, reducing the time required for an appointment and the assignment of external consultants and requiring a maximum delivery time of a medical–legal opinion within 6 months. Such abatement could bring the average closure time very close to 6 months, an impressive period, to say the least, for the handling of claims in the field of health, thus making the Conciliation Commission a unique management model for Europe in every respect. A very interesting opportunity offered by the Conciliation Commission, which is also unique in Europe, is the possibility of direct confrontation between the applicant and the practitioner, since in all the first hearings, and often also in subsequent meetings if possible, the presence of the professional involved in the event is strongly requested by both the commission and also by the healthcare company itself. Their presence offers the possibility of having an exchange of views between a patient and professional, allowing, in some cases, an agreement to be reached between the parties without acknowledging liability thanks to the explanations offered by the professional, explanations which may be meagre during the course of treatment and thus lead to misunderstandings and dissatisfaction. The possibility of having such an interview also makes it possible to rebuild a relationship of trust between the company itself and the patient, thus improving a patient’s opinion of the service offered by the provincial health system and promoting and facilitating possible future access to the services offered.

## Figures and Tables

**Figure 1 healthcare-12-01070-f001:**
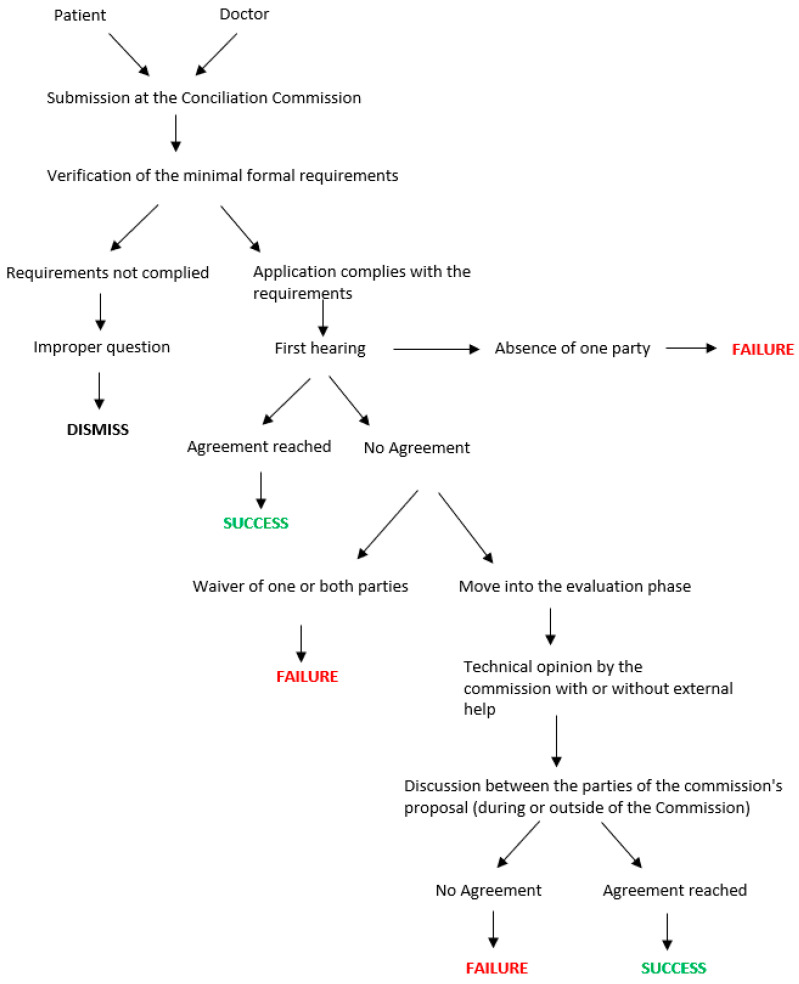
Scheme of Conciliation Commission procedure.

**Figure 2 healthcare-12-01070-f002:**
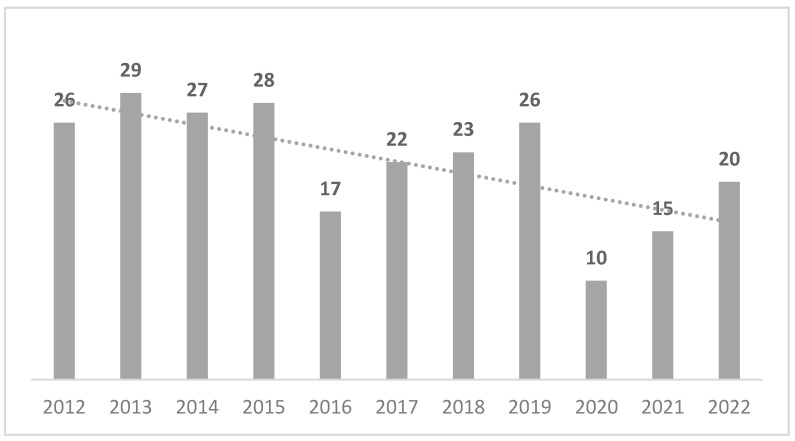
Time trend in the submission of applications to the Conciliation Commission from 2012 to 2022.

**Figure 3 healthcare-12-01070-f003:**
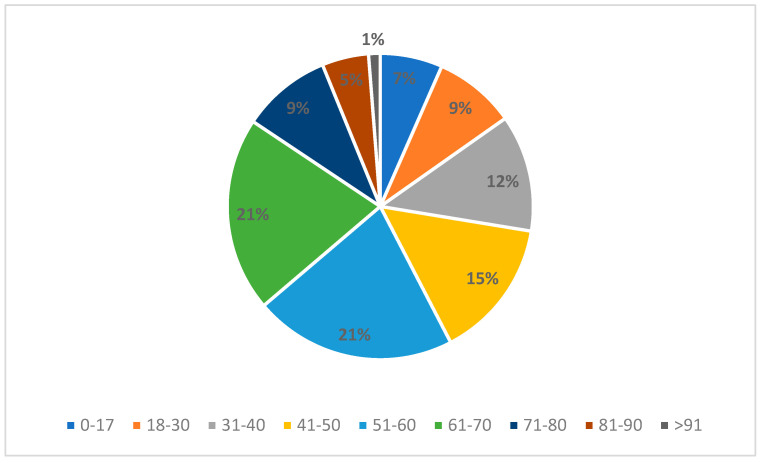
Age distribution of the injured party.

**Figure 4 healthcare-12-01070-f004:**
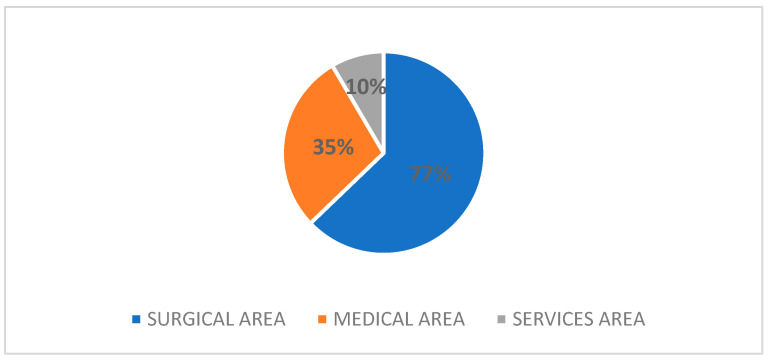
Type of health areas involved in Conciliation Commission claims.

**Figure 5 healthcare-12-01070-f005:**
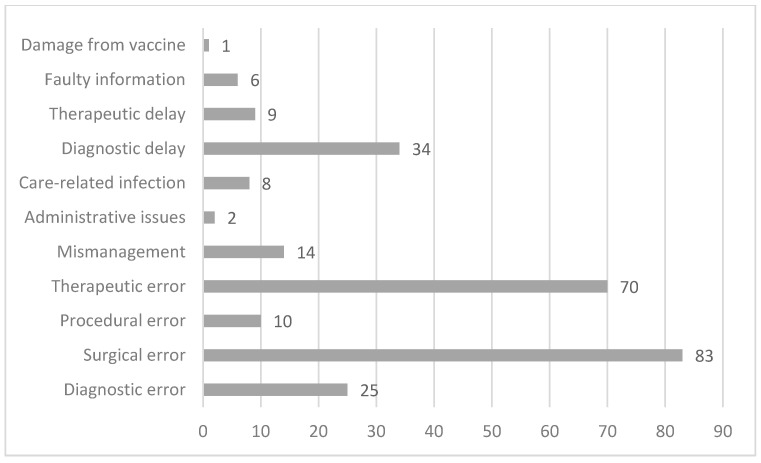
Type of complaint made by the applicant in the application form.

**Figure 6 healthcare-12-01070-f006:**
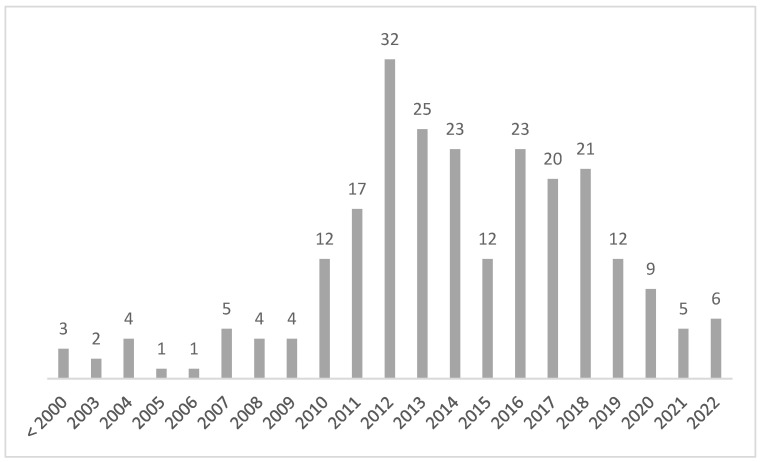
Time distribution of events of the claims from 2012 to 2022.

**Figure 7 healthcare-12-01070-f007:**
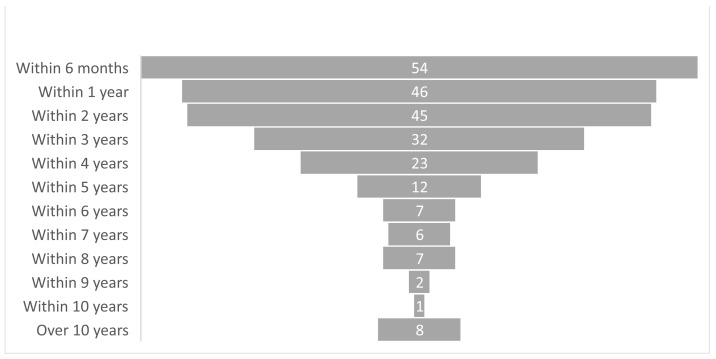
Time distance in months and years from the date of the event to the date of application.

**Figure 8 healthcare-12-01070-f008:**
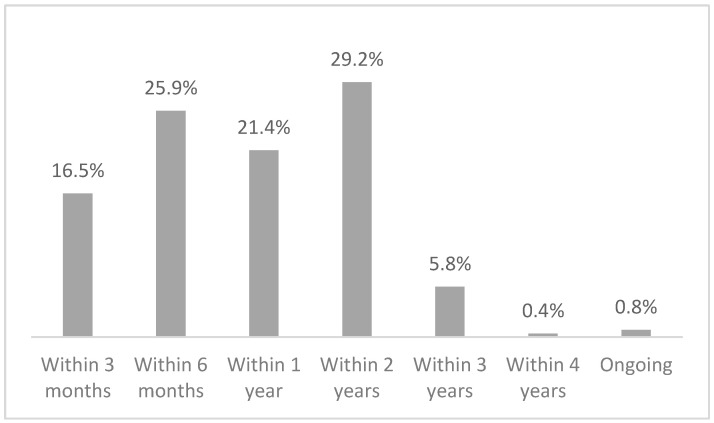
Time gap between the date of submission of an application and the date of closure of the procedure in the Conciliation Commission.

**Figure 9 healthcare-12-01070-f009:**
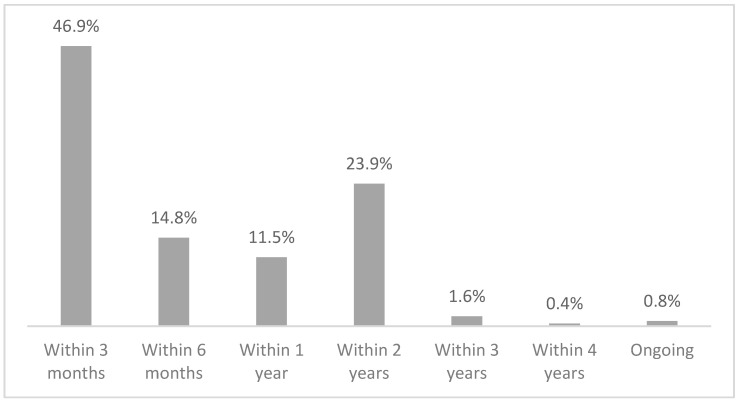
Duration of the commission’s operations from the first hearing to the closure date.

**Figure 10 healthcare-12-01070-f010:**
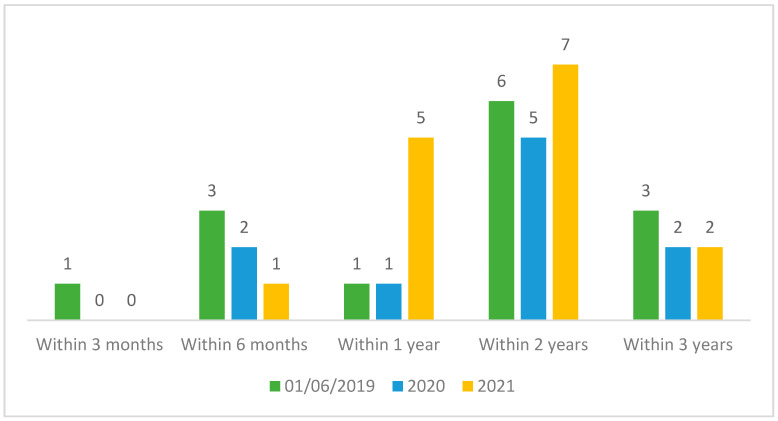
Time gap between an application’s date and closure date of the procedure in the Conciliation Commission during the pandemic period.

**Figure 11 healthcare-12-01070-f011:**
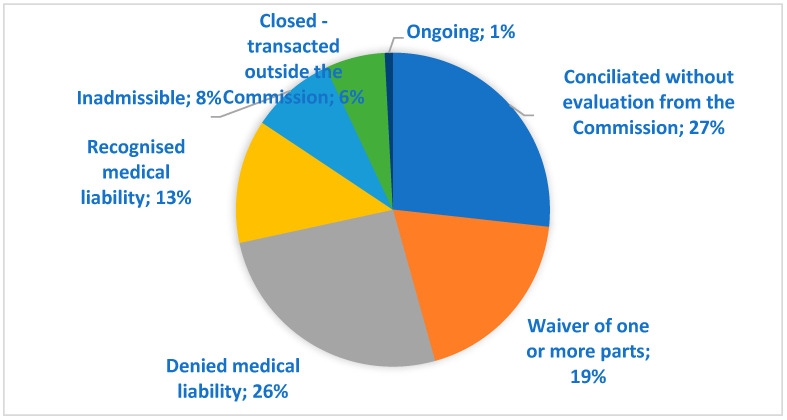
Conclusion of the Conciliatory Commission by 31 January 2024.

**Figure 12 healthcare-12-01070-f012:**
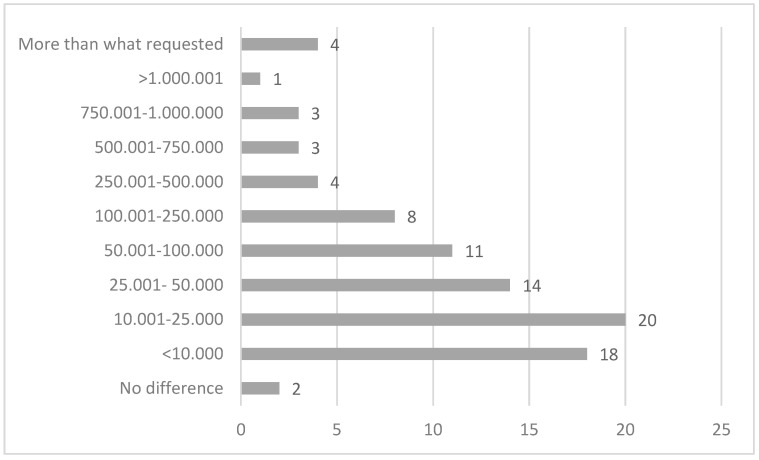
Difference between the sum requested in the application form and the sum suggested by the commission.

**Figure 13 healthcare-12-01070-f013:**
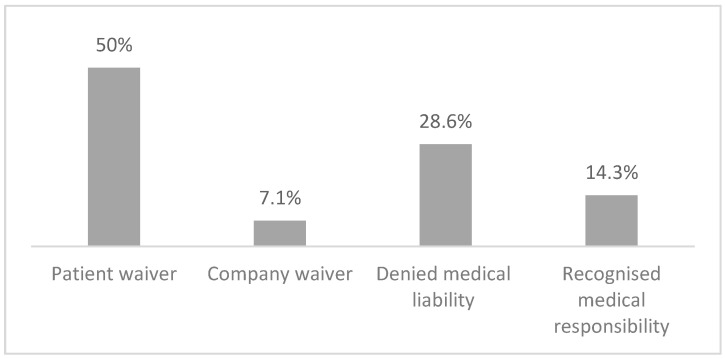
Conclusion of the commission in cases subsequently brought into dispute.

## Data Availability

The raw data supporting the conclusions of this article will be made available by the authors on request.

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
