# Peer review of "South Tyrol Healthcare Company: A Look at a Peculiar Model of Claims Management in Italy and Analysis of Its Last 11 Years"

_healthcare, 2024, doi:10.3390/healthcare12111070_

Round 1
Reviewer 1 Report
Comments and Suggestions for Authors
The analyses of the systems for reaching out-of-court settlements when a claim for medical assistance is presented are very interesting. The article presents the system developed by South Tyrol Healthcare Company and the results obtained between 2012-2022.
I am sending the authors some comments and suggestions:
Regarding the Introduction
It is proposed to clarify some of the terms of the study. For example, is the South Tyrol health company an insurance company or a public institution, or…?
It would also be important to define what is considered a medial claim or the type of claims that are presented to the Conciliation Commission. And its relationship with medical errors.
On several occasions it is said that the system has zero cost for the Health Company. It is necessary to explain who was responsible for the costs of developing the procedure and compensation.
The objective of the study should be redefined: “The aim of this study is to evaluate the effectiveness of the Conciliation Commission as an ADR through a comparison with other ADR systems in Europe and available data on the judicial handling of health liability cases in Italy.” After reading the results and the Discussion, it seems that the aim of the study is to present the system implemented by the South Tyrol Healthcare Company. Which is itself interesting.
And comment, not so much evaluate, its effectiveness comparing it with other ADR systems in Italy (judicial data are provided, not from other ADR systems) or other European countries that do not have similar systems.
Regarding the Results:
It is not clear who pays compensation for claims or the role of insurance companies.
If possible, it would be good to explain why patients or professionals or institutions abandon the negotiation or conciliation procedure.
It would be interesting to have a better explanation of how compensations are calculated, because they generally seem like low amounts.
Would it be possible to know how the existence of the ADR system for the extrajudicial resolution of cases is spread among professionals or patients?
Can any patient access the system, regardless of whether they have been treated in public or private institutions?
Regarding the Discussion
In this section, in addition to commenting on the results of the application of ADR, it is compared with some judicial data from Italy, which cannot be homologated.
It also refers to programs or institutions in other European countries. But some of these institutions analyse and compensate for objective harm caused to patients, while in the case of Bolzano it seems that this is not the case, since liability is also assessed. This is an important issue and must be considered because it makes systems comparable or not.
In other cases, it refers to systems that involve professional associations, which must also be differentiated since legal responsibilities differ.
Regarding the Conclusions
The conclusion insists that the system presented is unique in Europe, but there are mediation systems in other countries (European and North American) and this statement seems risky.
Similarly, I would remove this sentence from the Abstract: "the system is uniqueness in the world". There are systems of mediation and out-of-court dispute resolution in the various countries.
And also, I propose to omit the word ‘unique’ from the title.
Finkelstein A, Brezis M, Taub A, Arad D. Disclosure following a medical error: lessons learned from a national initiative of workshops with patients, healthcare teams, and executives. Isr J Health Policy Res. 2024 Mar 11;13(1):13. doi: 10.1186/s13584-024-00599-8. PMID: 38462624; PMCID: PMC10926562.
Manaouil C, Rahal-Löfskog D, Montpellier D, Jardé O. Fonctionnement pratique des commissions régionales de conciliation et d'indemnisation [Regional conciliation and compensation commissions: how they work]. Presse Med. 2006 Nov;35(11 Pt 2):1707-1715. French. doi: 10.1016/S0755-4982(06)74885-0. PMID: 17086130.
Manaouil C, Rahal-Löfskog D, Montpellier D, Jardé O. Fonctionnement pratique des commissions régionales de conciliation et d'indemnisation [Regional conciliation and compensation commissions: how they work]. Presse Med. 2006 Nov;35(11 Pt 2):1707-1715. French. doi: 10.1016/S0755-4982(06)74885-0. PMID: 17086130.
Regarding the Keywords: If you consider it, you could add a keyword, such as Medical Claims or Medical Claim Disclosure, Conciliation and compensation in Medical Claims
In summary, professionals, regulators, hospitals, accrediting organizations, and policymakers have been around for years and are taking steps to close the gap by developing standards, programs, and laws that encourage transparent communication with patients after harmful mistakes have been made. And establish modes of resolution without the need for litigation. It is an important benefit for everyone: patients and their families, health care providers and professionals, and society as a whole. Hence, an article that can provide experiences on how to do this is welcome.
Author Response
The analyses of the systems for reaching out-of-court settlements when a claim for medical assistance is presented are very interesting. The article presents the system developed by South Tyrol Healthcare Company and the results obtained between 2012-2022.
I am sending the authors some comments and suggestions:
Regarding the Introduction
It is proposed to clarify some of the terms of the study. For example, is the South Tyrol health company an insurance company or a public institution, or…? à Fair point. The information has been added within the abstract
It would also be important to define what is considered a medial claim or the type of claims that are presented to the Conciliation Commission. And its relationship with medical errors. à the term medical claim is never used in the article and the claims presented to the Conciliation Commission are those regarding the cases highlighted from line 86 to 89
On several occasions it is said that the system has zero cost for the Health Company. It is necessary to explain who was responsible for the costs of developing the procedure and compensation. à line 126 states that the costs of the procedure in the commission are due to the Public Italian Administration. Compensations costs are charged to the insurance company of the healthcare Company (line 114-115).
The objective of the study should be redefined: “The aim of this study is to evaluate the effectiveness of the Conciliation Commission as an ADR through a comparison with other ADR systems in Europe and available data on the judicial handling of health liability cases in Italy.” After reading the results and the Discussion, it seems that the aim of the study is to present the system implemented by the South Tyrol Healthcare Company. Which is itself interesting. And comment, not so much evaluate, its effectiveness comparing it with other ADR systems in Italy (judicial data are provided, not from other ADR systems) or other European countries that do not have similar systems. à Fair point. The aim of the study was to present the commission, but it also gives an idea of its effectiveness through the comparison with other ADR systems in Europe (line 427-464) with the limitations given by the available data and the way they are presented in literature. Thus we I think it is not wrong to include this aspect in the aim. Data on ADR systems in Italy is almost non-existent and what is available is about civil litigation in its entirety, without any distinction for medical liability cases, so the data in the authors' opinion cannot be compared. Therefore the aim was changed to “The aim of this study is to present the system of the Conciliation Commission implemented in South Tyrol and also to evaluate the effectiveness of the Conciliation Commission as an ADR through a comparison with other analogue ADR systems in Europe and available data on the judicial handling of health liability cases in Italy”
Regarding the Results:
It is not clear who pays compensation for claims or the role of insurance companies. à line 126 states that the compensation is due to the insurance company, whose role is to follow up on the proceedings and sometimes to propose a settlement to help bring closure to the case.
If possible, it would be good to explain why patients or professionals or institutions abandon the negotiation or conciliation procedure. à sadly this data is not available, because the one abandoning the conciliation procedure has no has no obligation to provide an explanation
It would be interesting to have a better explanation of how compensations are calculated, because they generally seem like low amounts. à The estimate is made based on the same method used in Italian judicial system in cases of health professional liability. This topic would open the door to a very complex discourse on the method of assessing the damage and its monetisation, which is beyond the scope of this article, so it has not been taken into consideration.
Would it be possible to know how the existence of the ADR system for the extrajudicial resolution of cases is spread among professionals or patients? à the conciliation commission, which is established by law and is part of the services offered by the public administration, has its own webpage.
Can any patient access the system, regardless of whether they have been treated in public or private institutions? à Fair point. The information has been added within the introduction.
Regarding the Discussion
In this section, in addition to commenting on the results of the application of ADR, it is compared with some judicial data from Italy, which cannot be homologated.
It also refers to programs or institutions in other European countries. But some of these institutions analyse and compensate for objective harm caused to patients, while in the case of Bolzano it seems that this is not the case, since liability is also assessed. This is an important issue and must be considered because it makes systems comparable or not.
In other cases, it refers to systems that involve professional associations, which must also be differentiated since legal responsibilities differ. à a new part about the limitations of our study has been added at the end of the discussion.
Regarding the Conclusions
The conclusion insists that the system presented is unique in Europe, but there are mediation systems in other countries (European and North American) and this statement seems risky.
Similarly, I would remove this sentence from the Abstract: "the system is uniqueness in the world". There are systems of mediation and out-of-court dispute resolution in the various countries. And also, I propose to omit the word ‘unique’ from the title.
Finkelstein A, Brezis M, Taub A, Arad D. Disclosure following a medical error: lessons learned from a national initiative of workshops with patients, healthcare teams, and executives. Isr J Health Policy Res. 2024 Mar 11;13(1):13. doi: 10.1186/s13584-024-00599-8. PMID: 38462624; PMCID: PMC10926562.
Manaouil C, Rahal-Löfskog D, Montpellier D, Jardé O. Fonctionnement pratique des commissions régionales de conciliation et d'indemnisation [Regional conciliation and compensation commissions: how they work]. Presse Med. 2006 Nov;35(11 Pt 2):1707-1715. French. doi: 10.1016/S0755-4982(06)74885-0. PMID: 17086130.
Manaouil C, Rahal-Löfskog D, Montpellier D, Jardé O. Fonctionnement pratique des commissions régionales de conciliation et d'indemnisation [Regional conciliation and compensation commissions: how they work]. Presse Med. 2006 Nov;35(11 Pt 2):1707-1715. French. doi: 10.1016/S0755-4982(06)74885-0. PMID: 17086130. à Fair point. The term unique in the title has been changed to peculiar and the part “ uniqueness in the world” has been removed from the abstract.
Regarding the Keywords: If you consider it, you could add a keyword, such as Medical Claims or Medical Claim Disclosure, Conciliation and compensation in Medical Claims à great advice, we implemented the keywords.
In summary, professionals, regulators, hospitals, accrediting organizations, and policymakers have been around for years and are taking steps to close the gap by developing standards, programs, and laws that encourage transparent communication with patients after harmful mistakes have been made. And establish modes of resolution without the need for litigation. It is an important benefit for everyone: patients and their families, health care providers and professionals, and society as a whole. Hence, an article that can provide experiences on how to do this is welcome.

Reviewer 2 Report
Comments and Suggestions for Authors
Dear authors,
Thank you for the opportunity to review the exciting paper “South Tyrol Healthcare Company: a look at a unique model of claims management in Italy and analysis of its last 11 years”, which presents a study on the effectiveness of the Conciliation Commission in the autonomous province of Bolzano for resolving healthcare disputes through alternative dispute resolution (ADR). The study highlights the rapid resolution times, low costs, and the need for increased awareness of this system in South Tyrol. The direction of the manuscript is clear, focusing on the methods, results, and conclusions regarding the effectiveness of this system in resolving healthcare disputes. The study design and approach seem appropriate for the research objectives, focusing on the collection and analysis of claims data to evaluate the performance of the ADR system. The first impression results in a well-structured article. A detailed assessment is given below:
The aim of the study is well-defined and consistent with the rest of the manuscript, which is to evaluate the effectiveness of the Conciliation Commission in comparison to other ADR systems. Nevertheless, a well-defined research question is missing.
The study design and methods are appropriate, involving the systematic collection of claims data from the Conciliation Commission. However, it is unclear to me how the data was accessed and who specifically analysed it. Possible biases in the sample are not explicitly addressed in the presented analyses. Furthermore, you did not include a discussion around ethical aspects, which should be included.
In this section, you should substantiate your central statements with an in-depth discussion using current international literature. Furthermore, a critical discussion of the limitations of their study is missing and should be added, as well as implications for future research activities in this particular area of healthcare.
SUMMARY: The study provides valuable insights into the functioning of the Conciliation Commission and its impact on dispute resolution in the healthcare sector. Key messages include the quick resolution times, the low costs and the need to increase awareness of this system in South Tyrol, but you could deepen the discussion by comparing your findings with existing research in the field and addressing any limitations or contradictory data. The main concerns centre on the lack of comparison with current research, the need for a discussion on the implications for future research and applications, and addressing limitations of the study and conflicting data.
Author Response
Dear authors,
Thank you for the opportunity to review the exciting paper “South Tyrol Healthcare Company: a look at a unique model of claims management in Italy and analysis of its last 11 years”, which presents a study on the effectiveness of the Conciliation Commission in the autonomous province of Bolzano for resolving healthcare disputes through alternative dispute resolution (ADR). The study highlights the rapid resolution times, low costs, and the need for increased awareness of this system in South Tyrol. The direction of the manuscript is clear, focusing on the methods, results, and conclusions regarding the effectiveness of this system in resolving healthcare disputes. The study design and approach seem appropriate for the research objectives, focusing on the collection and analysis of claims data to evaluate the performance of the ADR system. The first impression results in a well-structured article. A detailed assessment is given below:
The aim of the study is well-defined and consistent with the rest of the manuscript, which is to evaluate the effectiveness of the Conciliation Commission in comparison to other ADR systems. Nevertheless, a well-defined research question is missing. à The research question is exactly to evaluate the effectiveness of the instrument through an analysis of its results over the past 11 years, which is one of the aim of the study.
The study design and methods are appropriate, involving the systematic collection of claims data from the Conciliation Commission. However, it is unclear to me how the data was accessed and who specifically analysed it. à data were collected from a dedicated database of the health care company of the cases handled in the conciliation commission and from the database of conciliation commission itself.
Possible biases in the sample are not explicitly addressed in the presented analyses. à In the authors opinion there was no need to explicate sample biases because most of the data covers quantitative data for which there is no bias since most of the information is related to timing and characteristics such as department, age, counseling Y/N and such.
Furthermore, you did not include a discussion around ethical aspects, which should be included. àThe study, as it does not include any physical people, does not necessarily require a discussion of the ethical aspects, as it merely presents the results of a public body working in the assessment of medical damage and liability.
In this section, you should substantiate your central statements with an in-depth discussion using current international literature. à The discussion of the results has already been made through a comparison of data from current international literature. Unfortunately, data on ADR systems in Italy is almost non-existent and the existent data is about ADR systems used for any branch of civil litigation, not just health-care related cases.
Furthermore, a critical discussion of the limitations of their study is missing and should be added, as well as implications for future research activities in this particular area of healthcare. à a new part about the limitations of our study has been added at the end of the discussion.
SUMMARY: The study provides valuable insights into the functioning of the Conciliation Commission and its impact on dispute resolution in the healthcare sector. Key messages include the quick resolution times, the low costs and the need to increase awareness of this system in South Tyrol, but you could deepen the discussion by comparing your findings with existing research in the field and addressing any limitations or contradictory data. The main concerns centre on the lack of comparison with current research, the need for a discussion on the implications for future research and applications, and addressing limitations of the study and conflicting data.

Round 2
Reviewer 2 Report
Comments and Suggestions for Authors
Dear authors,
Thank you for the revision and the feedback. I have no more recommendations. The following paper may still be of interest.
Sisodiya, D. S., & Dwivedi, S. (2024). The Role of ADR in Resolving Disputes Related to Medical Negligence. International Journal of Law and Social Sciences, 9(1), 34–41. https://doi.org/10.60143/ijls.v9.i1.2023.82
Author Response
Please see the attachment. The suggested paper has been added.
